# A Proteomic Study of the Effect of N-acetylcysteine on the Regulation of Early Pregnancy in Goats

**DOI:** 10.3390/ani12182439

**Published:** 2022-09-15

**Authors:** Peifang Yang, Xiang Chen, Xingzhou Tian, Zhinan Zhou, Yan Zhang, Wen Tang, Kaibin Fu, Jiafu Zhao, Yong Ruan

**Affiliations:** 1Key Laboratory of Plateau Mountain Animal Genetics, Breeding and Reproduction of Ministry of Education, Guiyang 550025, China; 2Key Laboratory of Animal Genetics, Breeding and Reproduction of Guizhou Province, Guiyang 550025, China; 3College of Animal Science, Guizhou University, Guiyang 550025, China

**Keywords:** N-acetylcysteine, goats, early pregnancy, proteome, uterus

## Abstract

**Simple Summary:**

Early pregnancy regulation is an extremely complex process that is influenced by various factors. We previously mined the differentially expressed genes affected by N-acetyl-L-cysteine (NAC) in early pregnancy in goats via transcriptome sequencing. We found that NAC increased the number of lambs by affecting the immune pathway in ewes and enhancing antioxidation. Based on this, we here explored the effect of NAC on early pregnancy in goats at the protein level. The results showed a difference in the expression of uterine keratin and increases in the levels of antioxidant indices and hormones in doe serum.

**Abstract:**

Dietary supplementation with N-acetyl-L-cysteine (NAC) may support early pregnancy regulation and fertility in female animals. The purpose of this study was to investigate the effect of supplementation with 0.07% NAC on the expression of the uterine keratin gene and protein in Qianbei-pockmarked goats during early pregnancy using tandem mass spectrometry (TMT) relative quantitative proteomics. The results showed that there were significant differences in uterine keratin expression between the experimental group (NAC group) and the control group on day 35 of gestation. A total of 6271 proteins were identified, 6258 of which were quantified by mass spectrometry. There were 125 differentially expressed proteins (DEPs), including 47 upregulated and 78 downregulated proteins, in the NAC group. Bioinformatic analysis showed that these DEPs were mainly involved in the transport and biosynthesis of organic matter and were related to the binding of transition metal ions, DNA and proteins and the catalytic activity of enzymes. They were enriched in the Jak-STAT signalling pathway, RNA monitoring pathway, amino acid biosynthesis, steroid biosynthesis and other pathways that may affect the early pregnancy status of does through different pathways and thus influence early embryonic development. Immunohistochemistry, real-time quantitative PCR and Western blotting were used to verify the expression and localization of glial fibrillary acidic protein (GFAP) and pelota mRNA surveillance and ribosomal rescue factor (PELO) in uterine horn tissue. The results showed that both PELO and GFAP were localized to endometrial and stromal cells, consistent with the mass spectrometry data at the transcriptional and translational levels. Moreover, NAC supplementation increased the levels of the reproductive hormones follicle-stimulating hormone (FSH), luteinizing hormone (LH), oestradiol (E2), progesterone (P4), superoxide dismutase (SOD), glutamate peroxidase (GSH-Px) and nitric oxide (NO) in the serum of does. These findings provide new insight into the mechanism by which NAC regulates early pregnancy and embryonic development in goats.

## 1. Introduction

Reproductive ability is an important economic trait of goats, and improving the breeding capacity of goats is an important way to improve their breeding efficiency. The characteristics of early gestation and embryonic development are important factors affecting goat fecundity.

The uterus, a critical reproductive organ in female animals, plays a key role in early embryonic implantation, growth and development. The main biological functions of the uterus are to accept embryos and ensure an ideal developmental environment for them [1]. The establishment of pregnancy is a complex physiological process. A series of structural and functional changes must occur in the maternal uterus at the early stage of the establishment of pregnancy to enable the implantation of early embryos and the growth and development of the foetuses [2,3]. The steady state of the intrauterine environment is also an important condition for successful pregnancy and normal foetal growth and development [4]. Approximately 30% of early embryo loss occurs in the early stages of pregnancy, which also shows the importance of a balanced intrauterine environment and precise implantation [5,6]. Of course, good and balanced maternal nutrition is also necessary for the early growth and development of embryos [7]. The uterus is rich in nutrients, including glucose, amino acids and electrolytes, that support the growth and development of early embryos [8]. However, disturbances in the balance of oxidants and antioxidants in early pregnancy can result in severe damage to lipid membranes, proteins and DNA, which hinders the further development and growth of the embryo [9]. Therefore, it is worthwhile to study the effects of antioxidants on intrauterine homeostasis and early pregnancy regulation.

An effective method of increasing the kidding rate of pregnant does is adding anti-inflammatory and antioxidant substances to the diet. Previous research has indicated that it is feasible to use diet to regulate the state of antioxidant stress in animals [10,11]. Dietary supplementation with biological agents of appropriate nutritional value is one way to improve the antioxidant and nutritional status of pregnant goats [12,13,14,15]. N-acetyl-L-cysteine (NAC) is an acetylated variant of L-cysteine. As a common mercapto supplier, NAC plays important roles in antioxidative, anti-inflammatory and antiapoptotic processes [16,17]. In addition, NAC plays important pharmacological roles in scavenging free radicals, regulating cell metabolism, preventing DNA damage, and regulating gene expression and signal transduction [18,19]. As NAC is a precursor of glutathione (GSH), supplementation with NAC can promote the synthesis of GSH in mice, increase intracellular ATP levels and delay cell senescence. NAC can also improve the quality of fertilized eggs and the development of early embryos and blastocysts in mice [20,21,22]. Furthermore, NAC can prolong the storage period of two-cell embryos, restore ovarian and uterine functions and promote the implantation of early embryos [23]. In many studies, NAC has been found to be a treatment for endometrial inflammation [24]. Similarly, studies on goat ovarian granulosa cells have found that NAC can suppress the ROS signal transduction pathway of apoptosis and prevent granulosa cell apoptosis and follicular atresia [25]. In testicular germ cells, NAC can reduce intracellular lipid peroxidation and DNA fragmentation, which has certain therapeutic potential for male sterility [26]. In short, NAC plays important roles in early pregnancy regulation, embryonic growth and development, and animal fertility improvement.

Through RNA-seq, we previously found that adding different concentrations of NAC to the diet of does positively affected the survival of goat embryos in early pregnancy and increased the number of lambs born. NAC supplementation (0.07%) significantly improved the reproductive performance of does by affecting the expression of genes involved in the endometrial anti-inflammatory pathway [27]. However, that study was performed only at the gene level, so the regulatory effect of NAC on early embryonic development at the protein level remains to be studied.

Tandem mass spectrometry labelling (TMT) combined with liquid chromatography–tandem mass spectrometry (UHPLC–MS/MSLC–MS/MS) is becoming a mainstream analytical method due to its advantages of high throughput, high sensitivity, high accuracy and high stability. This method has attracted considerable attention in various research fields [28]. For example, it has become a powerful tool with which to explore biological processes such as animal growth and development and disease occurrence and has been widely used in research on the regulatory mechanisms of animal production traits. Its advantage is that it can be used to analyse molecular regulatory mechanisms at more microscopic levels than other methods [29]. In our previous study, RNA-seq was used to analyse the changes in uterine transcript levels, and candidate genes that NAC affects in early pregnancy regulation in does were screened [27]. Based on these findings, the current study used mass spectrometry to explore the uterine proteomic characteristics during NAC treatment in early gestation. The objective was to identify proteins and signalling pathways affected by NAC in the regulation of early pregnancy and embryonic development in does and to further analyse the effect and mechanism of action of NAC supplementation on uterine keratin expression and embryonic development in early pregnancy.

## 2. Materials and Methods

### 2.1. Establishment of Early Pregnancy and NAC Feeding

The test animals were Qianbei-pockmarked goats (N = 60, aged 36 months) provided by Fuxing Husbandry Co., Ltd. (Guizhou, China) (106.198244 E, 28.26403 N). Before the experiment, these Qianbei-pockmarked goats were housed under conditions of constant temperature (25 °C) and a fixed light/dark cycle (12 h/12 h) at the site of Fuxing Husbandry Co., Ltd., so that all test does could achieve as similar a physiological state as possible. After the does adapted to the environment, the vagina of each doe was implanted with a fluorogestone acetate vaginal sponge plug (Sansheng Biotechnology, NingBo, China) for 12 day. After the sponge suppository was removed, equine chorionic gonadotropin (eCG, 330 IU/each) (Sansheng, NingBo, China) and prostaglandin (PG, 1 ml/each) (Sansheng Biotechnology, NingBo, China) were injected subcutaneously so that all does entered oestrus at the same time and released oocytes. After 48 h of synchronized doe oestrus, the first artificial insemination was performed with fresh semen from fertile bucks, and the second artificial insemination was performed after 12 h. The time of completion of the second artificial insemination was recorded as day 0. Sixty does were randomly divided into an experimental group (NAC, N1 = 30) and a control group (control, N2 = 30). The NAC group was fed a basal diet supplemented with 0.07% NAC (Fanhai Biotechnology Co., Ltd., Guangdong, China), while the control group was fed the basal diet only (Appendix A. for raw materials and nutrients). The experimental does were fed twice a day at 10 a.m. and 5 p.m. All does were given free access to water and were fed in this manner for 35 consecutive days.

### 2.2. Sample Collection and Processing

On days 0, 3, 7, 21, 30, and 35, blood was collected from the jugular veins of three does in the NAC and control groups using a collection vessel (Solarbio, Beijing, China) before morning feeding. The sera were isolated and measured for reproductive hormones (including progesterone (P4), oestradiol (E2), follicle-stimulating hormone (FSH) and luteinizing hormone (LH)) and biochemical indices (superoxide dismutase (SOD), glutamate peroxidase (GSH-PX) and nitric oxide (NO)). Serum hormones and biochemical indicators were detected according to the instructions of the corresponding test kits (Solarbio, Beijing, China). Six goats per group were slaughtered on the 35th day of feeding, and the left uterine horn tissues of 6 pregnant does in the NAC group and control group were collected. The left uterine horn tissue was rinsed with sterile phosphate buffer solution (PBS) (Solarbio, Beijing, China). Immediately after grouping, a portion was stored in liquid nitrogen in cryopreservation tubes and sent to Novogene Biotechnology Co., Ltd. (Beijing, China), on dry ice for proteome sequencing. The rest was immediately subjected to RNA and protein extraction. The right uterine horn was placed in 4% paraformaldehyde solution at room temperature in the dark for immunohistochemical testing.

### 2.3. Proteomics

Proteomic analysis was performed on six biologically duplicated uterine horn (top) samples from the NAC group and the control group.

#### 2.3.1. Protein Extraction and Digestion

A Reference Protein Extraction Kit (Solarbio, Beijing, China) was used to extract protein from 12 samples of uterine horn tissue. The BSA standard protein solution was prepared according to the instructions of a Bradford Protein Quantification Kit (Servicebio, Wuhan, China). The absorbance of the standard protein solution was used to draw a standard curve and calculate the protein concentration of the sample to be tested. Then, trypsin digestion, TMT labelling, HPLC separation and liquid chromatographytandem mass spectrometry (LC–MS/MS) were performed as described previously [30,31]. According to the requirements of the experimental protocol, 100 μL of 0.1 M TEAB buffer was added to redissolve the sample, and then 41 µL of acetonitrile-dissolved TMT labelling reagent (Thermo, Waltham, MA, USA) was added. Twelve uterine horn samples were differentiated (the control samples were labelled 126, 127C, 127N, 128C, 128N and 129C; the NAC samples were labelled 129N, 130C, 131C, 132C, 133C and 134N). The reactions were mixed with inversion for 2 h at room temperature, and then ammonia (final concentration 8%) was added to terminate the reaction. The labelled samples with equal volumes were mixed and lyophilized after desalting.

#### 2.3.2. LC–MS/MS Analysis and Database Search

After elution and classification, the fractions were separated using an Easy-NLC 1000 UPLC system (Thermo Science). A Q Exactive^TM^ HF-X mass spectrometer with a Nanospray Flex™ (ESI) ion source was used for secondary mass spectrometry to generate raw mass spectrometry data (.raw). To improve the quality of the results, PD2.4 software was used to further filter the retrieved results. Peptide spectrum matches (PSMs) with a confidence level of 99% or above and at least one unique peptide (unique peptide) were considered reliable PSMs. Only the trusted peptides and proteins were retained, and their false discovery rates (FDRs) were calculated. Those with FDRs greater than 1% were removed.

#### 2.3.3. Bioinformatic Analysis

OmicStudio (https://www.omicstudio.cn/tool, accessed on 8 January 2022) online service tools were used for bioinformatic analysis. In this study, *p* < 0.05 and a fold change (FC) >1.1 or a FC < 0.91 were used as the thresholds for screening differentially expressed proteins (DEPs). The proteins were functionally annotated and classified according to the Gene Ontology (GO) and Kyoto Encyclopedia of Genes and Genomes (KEGG) databases. GO enrichment analysis was performed according to three main categories: the Biological Process, Cellular Component, and Molecular Function categories. All the DEPs were mapped to the GO database to identify every associated term (http://www.geneontology.org/, accessed on 15 January 2022). The number of proteins per term was calculated, and a hypergeometric test was applied to find the GO entries that were significantly enriched for the DEPs compared with the all-protein background. KEGG analysis (http://www.genome.jp/kegg/pathway.html, accessed on 8 February 2022) was used to determine the differences in proteins that were significantly enriched, mainly with regard to their biochemical pathways and signal transduction pathways. The analysis and annotation of the GO and KEGG results were carried out using online service tools. Through bioinformatic analysis of the biological functions of the DEPs, we obtained theoretically evidence of how NAC affects early embryonic development in goats.

### 2.4. RNA Extraction and Real-Time Quantitative PCR (qRT–PCR) Analysis

Total RNA was extracted from uterine horn tissue in the NAC group and control group using TRIzol reagent (Solarbio, Beijing, China), and first-strand cDNA was synthesized using a StarScript II First-Strand cDNA Kit (GenStar, Beijing, China). The corresponding reaction conditions were 42 °C for 30 min and 85 °C for 5 min. The product was stored at −80 °C for later use. The expression levels of the genes GFAP, OLFML3, TOMM20, TUBA4A, PELO, SLC27A4, SLC2A1 and ITGAL in the uterine horn in the control group and NAC group were detected by qRT–PCR with SYBR dye (GenStar, Beijing, China).

β-Actin was selected as the reference gene, and the primers were synthesized by Sangon Bioengineering Co., Ltd. (Shanghai, China). The details of the qRT–PCR primers are shown in Appendix A. The qRT–PCR system was as follows (10 μL): 2× UltraSYBR mixture, 5 µL; positive and reverse primers (pmol/µL), 0.75 µL; cDNA (ng/µL), 1 µL; and ddH_2_O, 2.5 µL. The qRT–PCR program was as follows: predenaturation at 95 °C for 105 s; 40 cycles of denaturation at 95 °C for 15 s, annealing at 57 °C for 15 s, and extension at 68 °C for 30 s; and denaturation at 95 °C for 5 s. The final melting curve was created by heating in 0.5 °C steps from 60 °C to 95 °C, and the fluorescence acquisition time was 5 s. Each sample was run in triplicate, and the mRNA expression levels of the above genes in the uterine horn tissues of experimental does were detected with a CFX 9600 real-time PCR machine (Bio-Rad, Hercules, CA, USA). The raw data from the machine were calibrated and normalized against those of β-actin, and the mRNA expression levels were computed with the 2^−^^△△^^CT^ relative quantitation method [32].

### 2.5. Immunohistochemistry

After slaughter, the right uterine horn of each doe was preserved in 4% paraformaldehyde until immunohistochemical analysis was performed according to the kit’s instructions (Servicebio, Wuhan, China).

Immunohistochemistry was performed as described previously [33,34]. In short, paraffin sections of 5-μm thickness were attached to microscope slides, heated at 65 °C for 2 h, dewaxed in xylene, and then rehydrated in a graded series of ethanol. Subsequently, these sections were placed in 0.1 mol/L sodium citrate buffer (pH 6.0), boiled in a microwave for 15 min, and then cooled to 37 °C. After rinsing with PBS, the activity of endogenous peroxidase was blocked by immersing the sections in 3% (mass fraction) hydrogen peroxide (H_2_O_2_) for 15 min. To block the nonspecific staining process, the sections were immersed in a blocking solution (5% (mass fraction) bull serum albumin (BSA) in TBSTw (including Tris-HCl, NaCl, and Tween 20) at 37 °C for 15 min. The sections were then immersed for 12 h at 4 °C with a rabbit anti-GFAP monoclonal antibody (1:20,000 v:v, AiFang Biological, AF300446, Hunan, China) or a rabbit anti-PELO polyclonal antibody (1:1000, Bioss, bs-7821R, Beijing, China) in TBSTw. After rinsing with PBS, the sections were incubated with a horseradish peroxidase (HRP)-labelled secondary antibody (A0208, Beyotime Biotechnology, Shanghai, China) at 37 °C for 2 h and then with diaminobenzidine for 15 min. The primary antibody was replaced with nonspecific rabbit immunoglobulin G (IgG) as the negative control (NC). Digital images of the immunohistochemically stained sections were captured through a three-light-source microscope (80i, Nikon, Japan).

### 2.6. Western Blot Analysis

The polyclonal GFAP antibody (AF300466) and monoclonal PELO antibody (BS-7821R) used in this study were purchased from Aifang Bio (Hunan, China) and Boaosen (Beijing, China), respectively. To verify the upregulated and downregulated protein expression, total protein was extracted using a total protein extraction kit (Solarbio, Beijing, China) and quantified using a BCA protein quantification kit (Solarbio, Beijing, China). Goat uterine keratin lysates (15 μg) from the NAC group and control group were treated with SDS–PAGE buffer, separated by SDS–PAGE, and transferred to PVDF membranes by electrophoresis. The membranes were placed into 5% skim milk powder diluted in Tris-buffered saline with Tween-20 (TBST) buffer, enclosed in a container, and incubated at 37 °C for 3 h. The membranes were then incubated with the primary antibody, either a rabbit anti-polyclonal GFAP antibody (1:20,000) or a rabbit anti-PELO monoclonal antibody (1:1000), at 4 °C. The membranes were incubated with a secondary antibody conjugated with HRP at 37 °C for 2 h. The HRP was then detected with an enhanced chemiluminescence detection system (Beyo ECL Star, Beyotime Biotechnology, P0018A, Shanghai, China). ImageJ (18.0) software was used to calculate the grey value. SPSS (26.0) software was used for one-way ANOVA to calculate the differential expression values [34].

### 2.7. Statistical Analysis

All variables were compared using one-way ANOVA for multiple groups (SPSS 26.0 software). The tissue expression of DEPs at the transcriptional and translational levels was analysed. Student’s *t* test was also carried out. The results are presented as the mean ± standard error of the mean (SEM) for the control and experimental groups. A *p* value < 0.05 was considered to indicate statistical significance.

## 3. Results

### 3.1. Proteomic Analysis

#### 3.1.1. Sample Inspection and Quality Control

To determine the effect of 0.07% NAC on the differential expression of proteins in the uterine horns of Qianbei-pockmarked goats in early pregnancy, the expression of uterine keratin was first compared between the NAC group and the control group with the TMT method. The protein quality of the samples was determined by SDS–PAGE (Appendix A). A total of 399,470 (121,130 matches, 30.32%) secondary spectra were identified by mass spectrometry, and 58,137 peptides were obtained. A total of 6271 proteins were identified by MS, of which 6258 were quantified (Table 1).

The specific polypeptides of these proteins were generally 7 to 27 amino acids long (Appendix A). With the techniques used in the current study, a greater number of peptides identified indicates greater reliability of the results. Therefore, the coverage of protein identification can reflect the overall accuracy of the identification results. The results showed that 61.57% of the identified peptide sequences covered more than 10%, while 41.13% of the identified peptide sequences covered more than 20% (Appendix A). The reliability of repeated-sample data was verified by principal component analysis (PCA) (Appendix A) and the coefficient of variation (CV) (Appendix A). The criteria of FC > 1.1 (or <0.91) and *p* < 0.05 were adopted to screen for DEPs. In total, 125 DEPs were found in the NAC group compared with the control group, and the details of each DEP are listed in Appendix A. Among the 125 DEPs, 47 were upregulated and 78 were downregulated (Figure 1). The expression profiles of the DEPs in the 12 samples are presented in the form of a heatmap (Figure 2).

#### 3.1.2. Bioinformatic Analysis

GO enrichment analysis (Figure 3) revealed that in the biological process category, the proteins were mainly associated with the protein phosphorylation, intracellular protein transport, redox process, amino acid transmembrane transport, metabolic process, translation, and steroid hormone synthesis signal transduction pathway terms. In the cellular component category, the proteins were mainly associated with the nucleus, membrane and ribosome terms. In the molecular function category, the proteins were mainly associated with the enzyme activity (including transferase, protein kinase, hydrolase, antioxidant enzyme activity), transition metal ion binding and sequence-specific DNA binding terms. KEGG pathway analysis (Figure 4) showed that the identified DEPs were involved in 93 pathways (Appendix A), and some enriched KEGG pathways were related to tight junctions, oxidative phosphorylation, Jak-STAT signalling, amino acid biosynthesis, steroid biosynthesis and other processes. These pathways suggest that NAC may play an indirect or direct role in early pregnancy regulation and early embryonic development in does.

### 3.2. qRT-PCR Analysis

Genetic information is stored in the genomes of organisms. mRNA transcription is the basic process of gene expression, while the protein product carries out the molecular function of a gene [35]. To verify that the changes in transcript levels were consistent with those in protein levels, we detected changes in the mRNA expression levels of the upregulated protein-coding genes PELO, SLC27A4, SLC2A1, and ITGAL and the downregulated protein-coding genes GFAP, OLFML3, TUBA4A, and TOMM20 in the uterine horns of does in the NAC and control groups by qRT–PCR (Figure 5). The mRNA expression of the upregulated genes was significantly higher in the NAC group than in the control group (*p* < 0.01) (A). In addition, the mRNA expression of the downregulated protein-encoding genes in uterine horn tissues of does was significantly lower in the NAC group than in the control group (*p* < 0.01) (B). The results were consistent with the mass spectrometry data.

### 3.3. Immunohistochemical Analysis

Immunohistochemistry was used to detect the expression and localization of GFAP and PELO proteins in the uterine horn tissues of goat does (Figure 6). Both GFAP (A) and PELO (B) were evidently expressed in the endometrium and stroma in goat uterine tissue, and NC reactions confirmed the absence of nonspecific staining for all target antibodies. These results suggest that GFAP and PELO may affect the reproductive performance of goats by regulating the function of endometrial cells.

### 3.4. Western Blot Analysis

To verify the tissue expression levels of the DEPs, we tested whether PELO was upregulated and GFAP was downregulated at the protein level. The results showed that the expression level of GFAP protein in vitro was significantly lower in the NAC group than in the control group (*p* < 0.01) and that the expression level of PELO protein in vitro was significantly higher in the NAC group than in the control group (*p* < 0.01) (Figure 7). The results were consistent with the in vivo proteomic data.

### 3.5. Analysis of Reproductive Hormones and Biochemical Indices in the Serum of Does

To investigate the effects of 0.07% NAC on the levels of reproductive hormones and biochemical indices in the serum of does in early pregnancy, the levels of FSH, LH, E2, P4, GSH-Px, SOD and NO in the serum of does at different times of gestation were detected. The results showed that there were no significant differences in reproductive hormone indices between the NAC group and the control group on day 0 of gestation (*p* > 0.05). On days 3, 7, 21, 30 and 35 of gestation, the levels of serum reproductive hormones were higher in the NAC group than in the control group (*p* < 0.05) (Table 2). During the trial phase, the reproductive hormone indices in the control group were relatively stable, while E2 and P4 in the NAC group showed an increasing trend, which may have underlain the oestrogen secretion in the luteum in pregnancy. These results suggest that NAC supplementation may affect the secretion of reproductive hormones in early gestation. There were no significant differences in serum biochemical indices between the NAC group and the control group on day 0 of gestation (*p* > 0.05), and the levels of serum biochemical indices in the NAC group were significantly higher than those in the control group on days 3, 7, 21, 30 and 35 of gestation (*p* < 0.05) (Table 3).

## 4. Discussion

The factors affecting early pregnancy regulation and embryonic implantation are important topics in the field of animal developmental biology [36]. In a previous study, we found that supplementation with 0.07% NAC in the diets of does in early pregnancy can improve the pregnancy rate, and we obtained the corresponding differentially expressed genes through RNA-seq [27]. Given our findings, it is of great significance to study the beneficial effects of NAC on early goat embryonic development in order to improve the lambing rate and fertility of does.

### 4.1. NAC Affects the Uterine Keratin Expression Profiles of Does in Early Pregnancy

In addition to the ovaries, the uterus plays a crucial role in female reproduction, including throughout the development process [37]. NAC is a widely used sulfhydryl compound that acts directly or indirectly on placental trophoblast (pTr) cells in a dose-dependent manner and may regulate placental function by regulating pTr steroid synthesis, cell proliferation and apoptosis, affecting pTr differentiation [38]. Some studies have shown that the amino acids required by pregnant does are devoted mainly to providing energy for foetal growth, promoting foetal protein deposition and maintaining normal maternal functions [39]. In this study, TMT quantitative proteomics was used to investigate the influence of 0.07% NAC supplementation on the differential expression of proteins in the uterine horn. A total of 6258 proteins were quantified in this study. In total, 125 DEPs were obtained in the NAC group compared with the control group, including 47 upregulated proteins and 78 downregulated proteins. Notably, our previous research has showed the effects of antioxidants on the growth performance, meat quality [40] and rumen fermentation parameters [41] of Qianbei-pockmarked goats. In the current study, bioinformatic analysis showed that the identified DEPs were mainly involved in the transport and biosynthesis of organic matter and were enriched with enzyme activity, transition metal ion and specific molecular binding function terms. The analysis of the related signalling pathways revealed that many of the DEPs were involved in the Jak-STAT signalling pathway, RNA monitoring pathway, tight junction pathway, and steroid hormone and amino acid biosynthesis pathway. Previous studies have shown that NAC supplementation during pregnancy can reduce oxidative stress and the inflammatory response of the maternal placenta and improve placental function, showing that NAC plays important roles in early embryonic development and in improving female fertility [42]. Therefore, the discovery of DEPs and functional pathways in this study provides basic data to help elucidate the mechanism by which NAC affects early pregnancy regulation and embryonic development in does.

### 4.2. Tissue Expression and Localization of DEPs

Earlier studies have found that GFAP is an astrocyte marker that is involved in the regulation of the division, renewal and proliferation of neural stem cells (NSCs) [43,44,45]. PELO, an evolutionarily conserved polymorphic protein, is related to the regulation of animal reproductive physiology. PELO not only participates in the regulation of various life activities [46] but also plays an important role in regulating cell migration and metastasis in vivo [47]. Other studies have found that PELO is involved in the regulation of mRNA translation [48] and that mutations in genes encoding PELO can lead to meiosis arrest of *Drosophila melanogaster* germ cells [49]. The results of the current study indicated that the protein levels of PELO and GFAP were altered after NAC feeding, which may have been related to the involvement of NAC in the biological processes of early pregnancy regulation and embryonic development in does. In this study, we found that GFAP was involved in the regulatory network of the Jak-STAT signalling pathway and that PELO was involved in the mRNA monitoring pathway. Immunohistochemistry showed that PELO and GFAP were localized in the stroma and intima cells of the uterus of the same doe. Western blotting showed that GFAP expression in the uterine horn of the NAC group was significantly downregulated and that PELO expression in the uterine horn of the NAC group was significantly upregulated. qRT–PCR showed that the upregulation of the protein-coding genes PELO, ITGAL, SLC27A4, and SLC2A1 and the downregulation of the protein-coding genes GFAP, OLFML3, TUBA4A, and TOMM20 at the transcriptional level were consistent with the expression levels of the corresponding proteins. Therefore, we hypothesized that 0.07% NAC supplementation affects the expression of GFAP and PELO at the transcriptional level and the translational level in the doe uterus and further participates in the regulation of early pregnancy and early embryonic development in does.

### 4.3. NAC Affects Reproductive Hormones and Biochemical Indices

The secretion and regulation of reproductive hormones guarantee the establishment of maternal pregnancy. According to the two-cell–two-gonadotropin theory, the growth and maturation of maternal follicles require the joint participation of FSH and LH [50]. LH and FSH are involved in the regulation of oestrogen release from the sow myolayer [51]. Moreover, FSHR in the placenta and myometrium can mediate the generation of blood vessels and regulate the contractile activity of the myometrium during pregnancy, activities that are beneficial for early embryonic implantation [52]. This study found that dietary supplementation with 0.07% NAC increased the serum levels of FSH and LH in does during early pregnancy and that the secretion of FSH oscillated, which is consistent with the findings of previous studies [53]. This process may be related to the establishment of gestation status in does. P4 is also important in the establishment and maintenance of pregnancy. P4 can stimulate the production and release of uterine secretions in early pregnancy, thus regulating pregnancy [54]. Oestrogen is a key hormone that controls maternal reproductive function, pregnancy establishment and early embryonic implantation [55]. Serum levels of oestrogen and progesterone are significantly increased in mice treated with antioxidants in early pregnancy [56]. In this study, E2 and P4 showed upward trends in early gestation, and the addition of NAC increased the levels of both of them, which is consistent with the results of previous studies. Antioxidant supplementation during pregnancy is beneficial for animals, playing important roles in improving blood antioxidant enzyme activity and maintaining cell integrity and normal function [57,58]. SOD and GSH-Px are important enzymes in the antioxidant defence system [59], effectively removing free radicals from the body and terminating free radical chain reactions [60]. We also found that NAC supplementation increased SOD, GSH-Px and NO levels in the serum of does in early pregnancy. These results indicate that NAC supplementation increases serum antioxidant levels in early-gestation does; thus, NAC may aid in gestation establishment.

Overall, NAC supplementation may benefit early pregnancy regulation in does, regulating reproductive hormones and biochemical indicators, and may therefore be conducive to establishing pregnancy.

## 5. Conclusions

Dietary supplementation with 0.07% NAC changed the expression profile of uterine keratin in early gestation. DEPs were mainly involved in the transport and biosynthesis of organic matter and were associated with enzyme activity, transition metal ions and molecular specific binding functions. KEGG pathway analysis showed that the DEPs were also involved in the Jak-STAT signalling, RNA monitoring, tight junction, steroid hormone and amino acid biosynthesis pathways. NAC also affected the secretion of reproductive hormones and the changes in biochemical indices in early gestation. These results regarding the effects of NAC on early pregnancy regulation provide theoretical and empirical bases for further analysis of the molecular mechanism by which NAC regulates pregnancy and influences early embryonic development.

## Figures and Tables

**Figure 1 animals-12-02439-f001:**
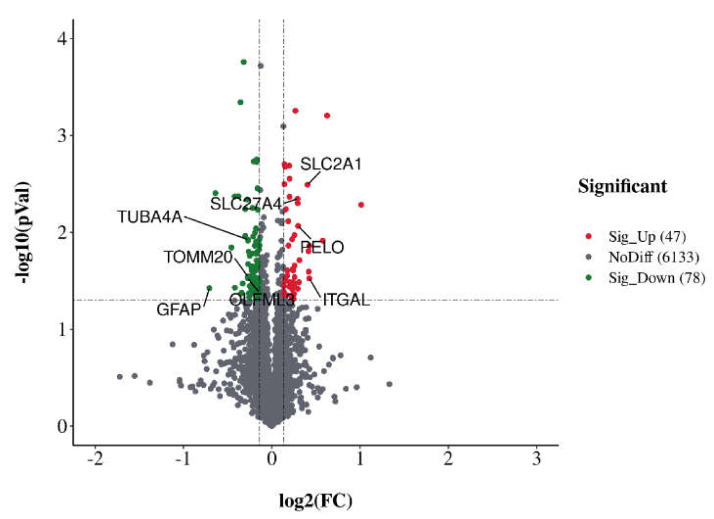
Volcano map of differentially expressed proteins. Proteins with fold change values >1.1 or <0.91 and *p* < 0.05 were considered to be significantly differentially expressed. Red circles indicate significantly upregulated proteins, green circles indicate significantly downregulated proteins and grey circles indicate proteins with no difference. The x-axis indicates the fold change, and the y-axis indicates the *p* value.

**Figure 2 animals-12-02439-f002:**
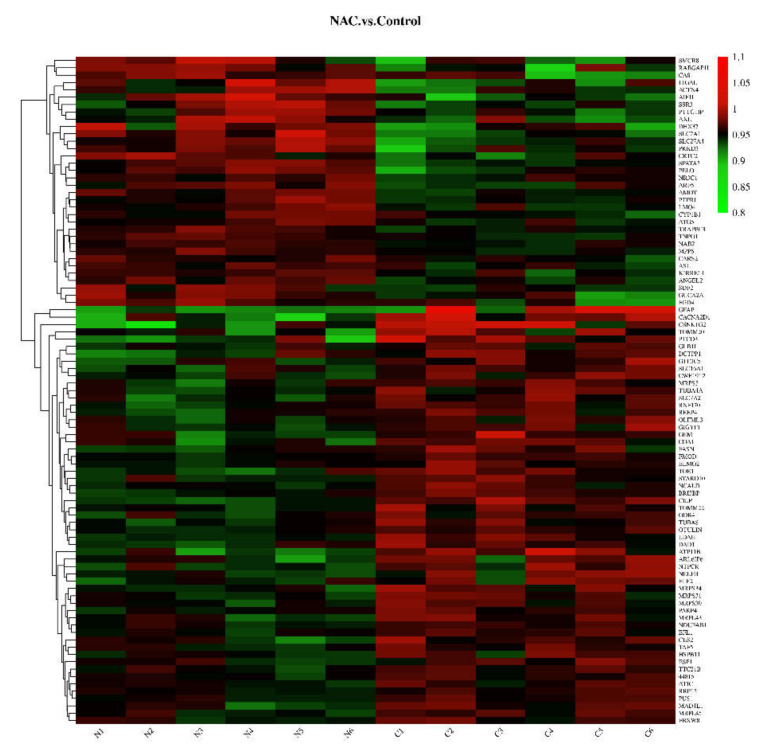
Clustering heatmap of differentially expressed proteins. N1, N2, N3, N4, N5, and N6, NAC group; C1, C2, C3, C4, C5, and C6, control group.

**Figure 3 animals-12-02439-f003:**
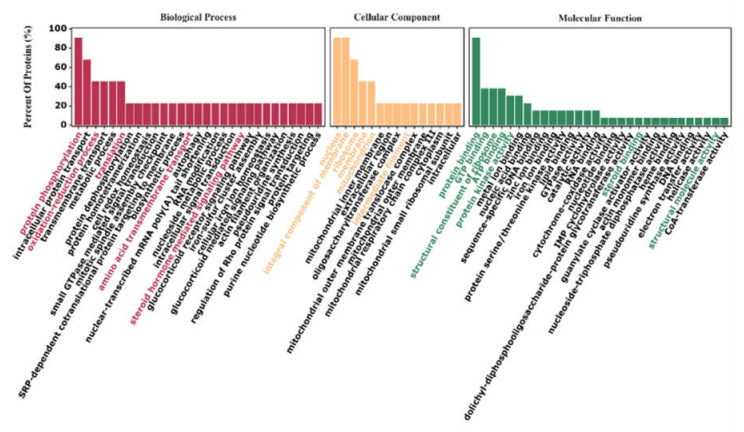
GO analysis of differentially expressed proteins (DEPs). Gene Ontology (GO) analysis of DEPs. The GO annotation categories included the Biological Process (BP) (red), Cellular Component (CC) (yellow), and Molecular Function (MF) (green) categories. The percentage of DEPs in each category is shown. Each category shows the top 30 GO terms with the greatest enrichment ratios.

**Figure 4 animals-12-02439-f004:**
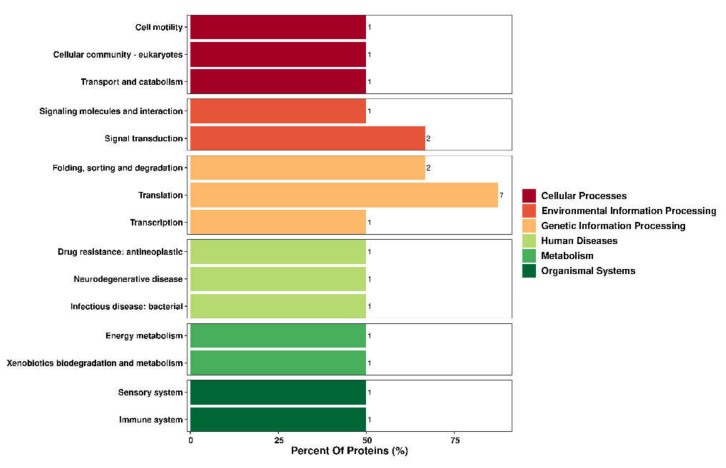
Kyoto Encyclopedia of Genes and Genomes (KEGG) annotation of differentially expressed proteins (DEPs). KEGG analysis of DEPs. The abscissa shows the percentage of DEPs, and the ordinate shows the the significantly enriched KEGG category.

**Figure 5 animals-12-02439-f005:**
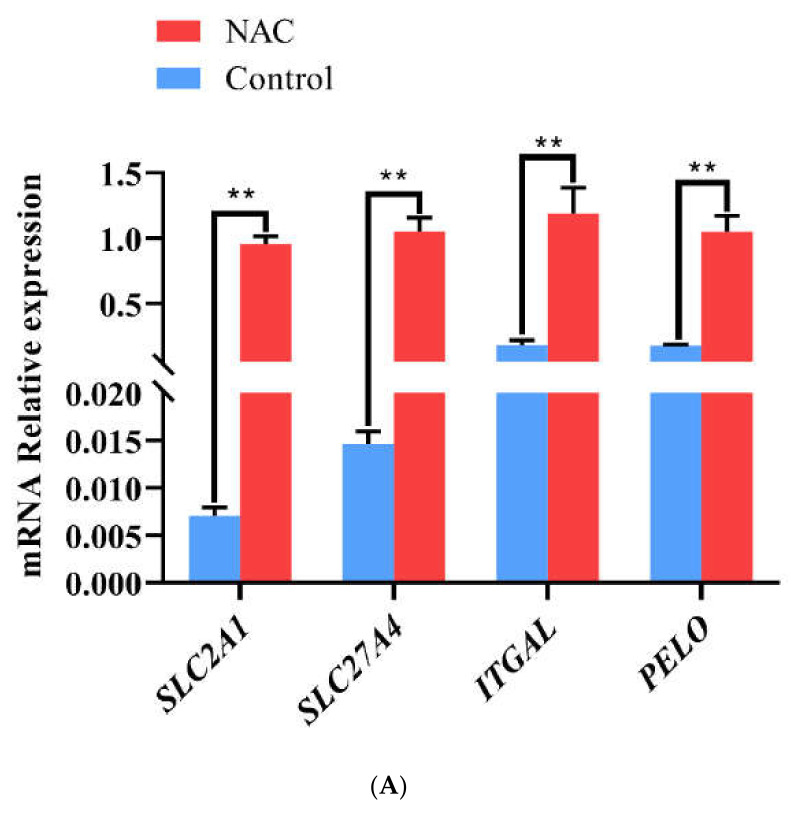
(**A**) Analysis of the transcript levels of upregulated differentially expressed proteins SLC2A1, solute carrier family 2 member 1; SLC27A4, solute carrier family 27 member 4; ITGAL, integrin, alpha L. ** indicates highly significant difference (*p* < 0.01). (**B**) Analysis of the transcript levels of downregulated differentially expressed proteins. *OLFML3*, olfactomedin-like 3; *TOMM20*, translocase of the outer mitochondrial membrane member 20; TUBA4A, tubulin alpha-4A. ** indicates highly significant difference (*p* < 0.01).

**Figure 6 animals-12-02439-f006:**
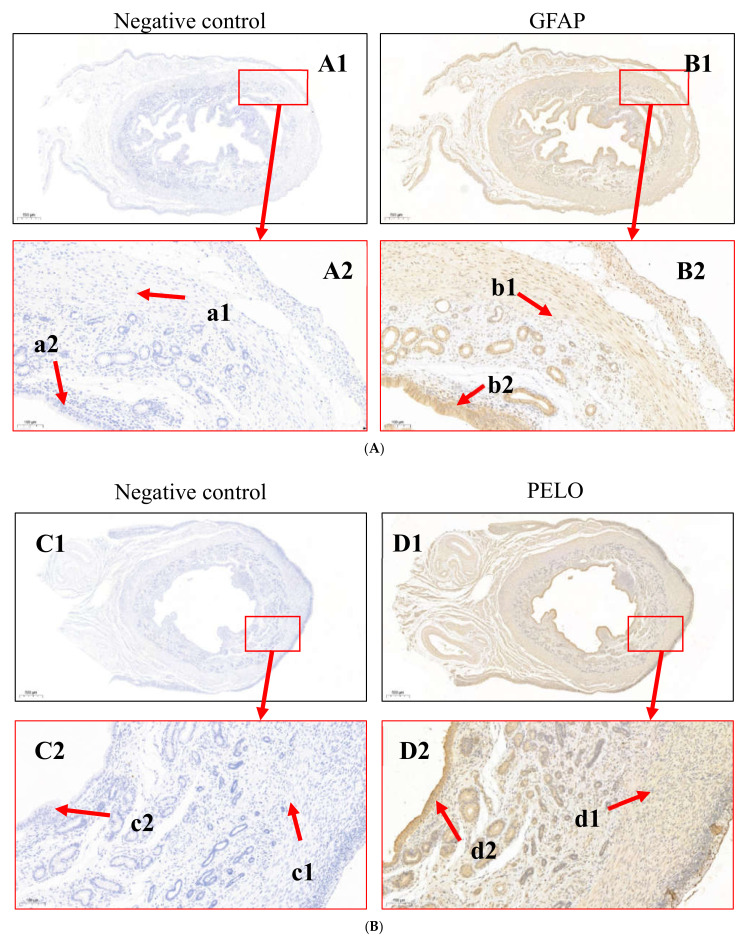
(**A**) Analysis of GFAP localization in the uterine tissue of Qianbei-pockmarked goats. a1 and b1 are uterine stromal cells. a2 and b2 are endometrial cells. (**B**) Analysis of PELO localization in the uterine tissue of Qianbei-pockmarked goats. c1 and d1 are uterine stromal cells. c2 and d2 are endometrial cells.

**Figure 7 animals-12-02439-f007:**
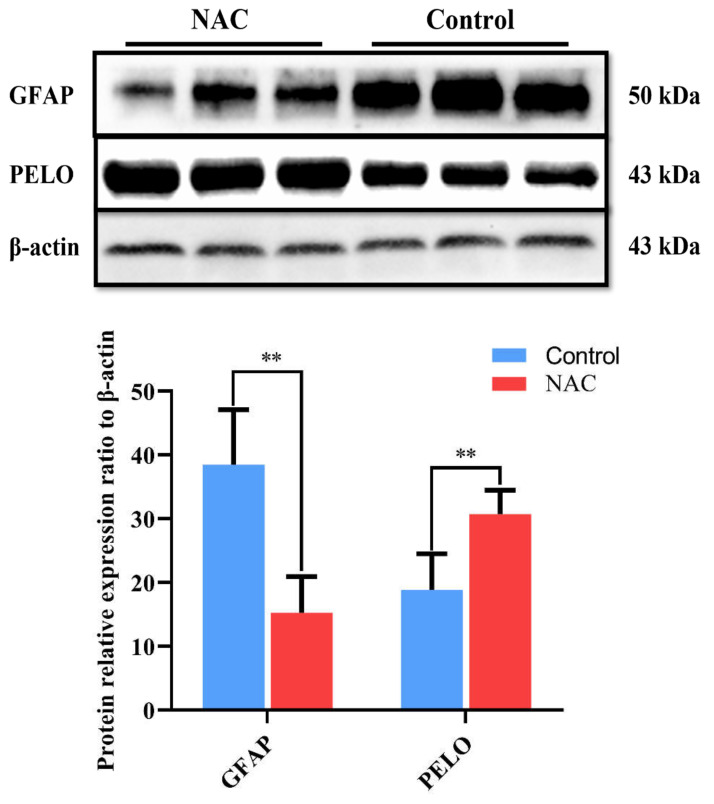
Effects of NAC on the protein expression of GFAP and PELO in the uteri of Qianbei-pockmarked goats. GFAP, glial fibrillary acidic protein; PELO, pelota mRNA surveillance and ribosomal rescue factor; ** indicates highly significant difference (*p* < 0.01).

**Table 1 animals-12-02439-t001:** Total numbers of identified peptides and proteins.

Run Name	Total Spectra	Matched Spectrum	Peptide	Identified Protein	All
NAC.vs.control	399,470	121,130	581,37	6281	6258

Total spectra: the total number of secondary spectra; matched spectra: the number of valid spectra; peptide: the number of identified peptides; identified protein: the number of identified proteins. ALL represents the total quantifiable proteins for all samples.

**Table 2 animals-12-02439-t002:** Effects of NAC on serum reproductive hormone indices in does at different times of gestation.

Item	Treatment	Different Gestation Times (Days)					
		0 d	3 d	7 d	21 d	30 d	35 d
E2 (ng/L)	NAC	57.69 ± 0.35 ^Ad^	71.17 ± 2.35 ^Ac^	70.42 ± 0.78 ^Ac^	83.75 ± 0.87 ^Aa^	77.95 ± 0.66 ^Ab^	82.59 ± 1.22 ^Aa^
	control	57.31 ± 0.20 ^Abc^	56.57 ± 1.75 ^Bc^	58.25 ± 0.66 ^Bbc^	59.99 ± 1.64 ^Bb^	63.53 ± 1.91 ^Ba^	60.52 ± 1.18 ^Bb^
FSH (IU/L)	NAC	28.48 ± 0.29 ^Ad^	33.08 ± 0.70 ^Ac^	34.25 ± 0.00 ^Ab^	32.29 ± 0.79 ^Ac^	34.28 ± 0.53 ^Ab^	35.67 ± 0.70 ^Aa^
	control	28.17 ± 0.43 ^Aabc^	28.61 ± 0.83 ^Ba^	27.20 ± 0.66 ^Bbc^	26.66 ± 0.01 ^Bc^	27.01 ± 0.46 ^Bc^	28.28 ± 0.22 ^Bab^
LH (ng/L)	NAC	24.46 ± 0.24 ^Ae^	27.80 ± 0.63 ^Ad^	30.60 ± 0.20 ^Ac^	32.66 ± 0.04 ^Ab^	34.58 ± 0.86 ^Aa^	34.12 ± 0.59 ^Aa^
	control	24.88 ± 0.41 ^Aab^	23.26 ± 0.48 ^Bb^	24.46 ± 0.79 ^Bab^	23.76 ± 0.68 ^Bb^	24.52 ± 0.20 ^Bab^	25.20 ± 0.84 ^Ba^
P4 (p mol/L)	NAC	1126.24 ± 34.27 ^Ad^	1257.04 ± 30.66 ^Ac^	1436.09 ± 34.72 ^Ab^	1521.63 ± 5.92 ^Aa^	1574.09 ± 18.85 ^Aa^	1535.32 ± 37.01 ^Aa^
	control	1149.83 ± 16.88 ^Aab^	1190.89 ± 12.95 ^Ba^	1168.08 ± 46.20 ^Bab^	1176.94 ± 21.04 ^Ba^	1113.59 ± 46.59 ^Bb^	1135.52 ± 20.92 ^Bab^

For the same index, different capital letters in the same column indicate significant differences between the NAC and control groups within the same gestation time (*p* < 0.05), and different lowercase letters in the same row indicate significant differences in the NAC group (or control group) between different gestation times (*p* < 0.05). E2, oestradiol; FSH, follicle-stimulating hormone; LH, luteinizing hormone; P4, progesterone.

**Table 3 animals-12-02439-t003:** Effects of NAC on serum biochemical parameters of does at different times of gestation.

Item	Treatment	Different Gestation Times (Days)					
		0 d	3 d	7 d	21 d	30 d	35 d
SOD (p g/mL)	NAC	243.43 ± 2.63 ^Ad^	266.41 ± 4.37 ^Ac^	275.72 ± 1.37 ^Aab^	268.93 ± 8.26 ^Abc^	277.77 ± 2.63 ^Aa^	265.31 ± 2.90 ^Ac^
	control	244.98 ± 3.78 ^Aab^	245.15 ± 6.71 ^Bab^	240.97 ± 2.88 ^Bb^	248.35 ± 0.89 ^Ba^	243.66 ± 1.51 ^Bab^	246.08 ± 0.94 ^Bab^
GSH-PX (p mol/mL)	NAC	57.93 ± 1.46 ^Ab^	63.12 ± 1.95 ^Aa^	63.38 ± 0.92 ^Aa^	62.45 ± 1.56 ^Aa^	61.54 ± 1.02 ^Aa^	62.50 ± 0.84 ^Aa^
	control	57.55 ± 1.80 ^Aa^	55.05 ± 2.50 ^Ba^	58.84 ± 0.36 ^Ba^	57.55 ± 0.98 ^Ba^	54.08 ± 0.65 ^Ba^	56.47 ± 5.28 ^Ba^
NO (μ mol/L)	NAC	34.60 ± 0.61 ^Ad^	43.84 ± 0.99 ^Ac^	44.37 ± 1.11 ^Ac^	43.80 ± 1.15 ^Ac^	47.97 ± 0.21 ^Aa^	46.42 ± 0.72 ^Ab^
	control	34.08 ± 1.05 ^Abc^	33.16 ± 0.53 ^Bc^	35.93 ± 1.13 ^Ba^	34.54 ± 1.05 ^Babc^	35.20 ± 0.95 ^Bab^	33.76 ± 0.44 ^Bbc^

For the same index, different capital letters in the same column indicate significant differences between the NAC and control groups within the same gestation time (*p* < 0.05), and different lowercase letters in the same row indicate significant differences in the NAC group (or control group) between different gestation times (*p* < 0.05). SOD, superoxide dismutase; GSH-PX, glutamate peroxidase; NO, nitric oxide.

## Data Availability

The raw data supporting the conclusions of this article will be made available by the authors, without undue reservation.

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
