# Peer review of "A Proteomic Study of the Effect of N-acetylcysteine on the Regulation of Early Pregnancy in Goats"

_animals, 2022, doi:10.3390/ani12182439_

Round 1
Reviewer 1 Report
The article seems well written, and the authors have done a great deal of work. However, the following minor edits are suggested.
- Pag 1; line 16, what is the P-value?
- Page 2; line 55, delete provide
- Page 2; line 81, replace lambing with kidding
- Page 3; line 109; how old were your female goats?
- Page 3; line 112, we normally acclimatize goats for 14 days to their diets
- Page 3; line 121, replace rams with bucks
- Page 3; line 124, what was the basal diet? What was the nutritional composition of the basal diet? How did you mix NAC into the diet?
- Page 3; line 131, replace neck veins with jugular veins
Author Response
I'm honored to have your suggestions on this manuscript. Corresponding amendments have been made to your suggestions. If there is anything wrong, I will continue to make amendments. In the attachment.

Reviewer 2 Report
Dear authors, it seems to me that this manuscript has great relevance in the scientific world. However, many points affect the quality of the manuscript. Please don't just answer me, correct the text.
Correct the language and writing style. Some lines are difficult to read.
The manuscript needs the simple summary. According to the “Instruction for authors”: Submissions without a simple summary will be returned directly.
The abstract is not complete, rewrite it. There is a lack of information and p-values. Remember, the abstract is the document that first represents your manuscript.
The introduction is very generic and with uncorrelated ideas. At some points it seems like a copy and paste of ideas. Rewrite it. Add numerical data to your introduction if possible. Add the hypothesis of the manuscript in the introduction.
The writing style of the results topic is very generic, rewrite it. Add the numbers found, percentages, rates, and other data to better understand your findings. Avoid write the table or figure title or similar.
The discussion is very generic, with many beliefs, which are ultimately just a scientific review. The topic of discussion is to explain biologically how you got these results and I didn't see these explanations in the discussion. Add those explanations. Explain your results.
The conclusion must be objective and direct. In this way it is a comment of the authors. Rewrite it.
Furthermore, I have problems with the term “biosynthesis of organic matter”. This is a general way to describe the item.
Lines 12-16: Rewrite the aim in an objective form and in an aim format. Rewrite it.
Lines 16-18: These lines are very generic. Describe if you saw effects: positive or negative; increasing or decreasing, etc. Rewrite it.
Lines 18-20: And what about the control group? This information is important to determine the treatment effects. Rewrite it.
Lines 22-25: Those lines are repetitive with lines 20-22. Rewrite it.
Lines 29-31: These lines are irrelevant. Remove it.
Lines 34-35: This is not a conclusion. According to the results, how does NAC act in early pregnancy and embryo development? Rewrite it.
Keywords: write “utero or uterus” instead of “the uterus”.
Lines 39-40: “traits … traits” … “breeding … breeding”. Repetitive. Rewrite it.
Lines 40-42: This is ok, but this idea has no correlation with the idea before or after. I'm talking about the writing style. Rewrite it.
Line 44: Remove the first “and”.
Line 45: Change “ideal” instead “good”
Lines 46-49: I don't know if it's the language or the writing style but this text is hard to read. Rewrite it.
Line 49: What do you mean or what are you trying to say by "balance"? Use another appropriate or technical word, or rewrite it.
Line 56: What do you mean with "good"? Use another appropriate or technical word. Rewrite it.
Line 61: What? At what point did antioxidant stress become a problem? There is a problem in writing style.
Line 62: What do you mean with "appropriate"? Rewrite it.
Lines 81-82: The ideal place for this idea is before line 61.
Lines 102-105: This is a well written aim. Use it to enhance the aim and conclusion of the abstract
Line 109: Add the average weight±SD and age±SDof the animals.
Lines 110-113: Please describe better what you mean by "same environment" and "production level". How much food and water did they receive? What antiparasitic drug and vaccines were used?
Line 114: Adapted to what? Describe better. Explain better the chronology of the events because it is confusing.
Line 119: Why do you use the word “selected”? You didn't use a sample of all 60 animals, you used all the animals.
Line 121: Were all the goats inseminated a second time?
Line 122: At what time were the goats randomly divided? After insemination? After adaptation?
Line 126: Add this supplemental table 1 into the manuscript as a normal table.
Lines 126-127: I don't understand this; what do you mean by this line?
Line 129: Starting when and ending when? Again, there are shortcomings in determining the chronology from reading the manuscript. Rewrite it and better describe the total chronology of the study.
Line 131: What veins? Please do not answer this in the letter, write it in the text of the manuscript.
Lines 138-139: Complete this methodology, was it an intrauterine collection? laparoscopy? Etc.
Lines 143-145: What? Was the uterus obtained after the slaughter of the goats? Or did the goats undergo a hysterectomy? It is not clear.
Line 147: What? Where in the uterine horn were the samples collected? Or was it the whole uterine horn?
Line 155: What is LS-MS/MS? Describe the complete name.
Line 157: How long?
Line 164: LC or LS (line 155)?
Lines 258-261: What is the point of these lines? I think these lines should be on the topic of material and method.
Lines 295-296: Remove it.
Line 368: Where was the in vitro methodology described?
Lines 372-374: These lines belong to the topic of discussion.
Lines 391-393: Remove it.
Lines 396-398: These lines belong to the topic of discussion.
Lines 440-442: How? Please don't answer me in the cover letter, correct the text.
Lines 449-452: How? Please don't answer me in the cover letter, correct the text.
Lines 466-467: What results? The results of the other studies or your study?
Author Response
我很荣幸能有你对这份手稿的建议。已对您的建议进行了相应的修改。如果有什么问题,我将继续进行修正。在附件中。

Reviewer 3 Report
The manuscript “Proteomic study on the effect of N-acetylcysteine on the regulation of early pregnancy in goats” analyses the addition of 0.07% of NAC to the diet of pregnant goats and its effect on the regulation of early pregnancy and embryo development. The research is well justified by the introduction, adequately written, and interesting. The materials and methods used are very complete in order to fulfill the aims of the research and have a good methodology but there are some corrections needed:
L24, 81: in these lines and the whole manuscript the term “doe goats” is used, erase “goats” and leave only “doe” or “does” (accordingly) it is unnecessary to specify they are goats.
L46: References should be between brackets [ ], not parenthesis.
L116: pregnant horse serum gonadotropin (pmsg) is an obsolete term, change for eCG, equine chorionic gonadotropin.
L120: what do you mean by “mentally healthy” please clarify
L131: which “neck vein”, do you mean the jugular vein? You should specify
L138: If the animals were slaughtered, you should specify the method.
L252: You should be more specific about the statistical analysis done for each variable, not just say “in some cases, …”
Regarding results and discussion, some changes are needed, and in my opinion, they need to be restructured, as they seem to be intermixed, and they should be separated. Also, in general, the figures’ legends need to be rewritten, any figure should be comprehensible on its own, without the need to read the main text. Some figures (i.e. figure 2) are too small to be able to understand, it is advisable to make them a little bigger. The discussion should be completely restructured because it should not include results (i.e. L446-449), and the possible explanation for the results obtained should be thoroughly discussed in this section, and not mentioned in the results section (i.e. L396-398). A more precise list of corrections is listed below:
L270-275: Remove the space between the title and the explanation of the figure, it is using unnecessary space, do this in any figure that has this issue.
L311, 317-319: Place the explanation for figure 3 below the figure, as the text for figures 3 and 4 gets mixed.
L338-344: It would be advisable to make this just one figure with an internal explanation for A and B, differentiating both graphics with the actual letters, and not separating them as if they were two figures. Could be placed one beside the other.
L352-364: the same as above, these two figures could be restructured to be just one with its differentiation, it is necessary that every letter placed in the figure is defined in the text (capital letters with their numbers are not defined)
L371-374, 389-391, 396-398: The results section should only mention results, not the possible reasons for the findings, which should be discussed in the discussion section.
L417-436: this paragraph seems more of a background and justification for the research than an actual discussion.
L442-449: As mentioned before, results are mixed with the discussion, they should be separated.
L507-508: Some given possible explanations for the findings are written without the necessary background and further explanation of why this is thought to be the reason for the findings.
Overall, I recommend that the discussion is restructured and rewritten in order to better explain the possible reasons for every finding, the research is very complete and interesting, but the discussion falls short and doesn’t make justice to the findings. The discussion should have an order and give an overall explanation of how every finding is related to the rest and why they are important.
Regarding the conclusion, it also needs to be restructured, it seems more of a summary of the findings and not an actual conclusion, which should be more like L509-511.
Author Response
我很荣幸能有你对这份手稿的建议。已对您的建议进行了相应的修改。如果有什么问题,我将继续进行修正。请参阅附件。

Round 2
Reviewer 2 Report
Dear authors, you do a great job improving the content of your manuscript. I think the manuscript is ready to be published.
Author Response
我已根据相应要求进行了更改。
